Inconsistent effects of components as evidence for non-compositionality in chimpanzee face-gesture combinations? A response to Oña et al (2019)

Cauté Maxime 1 maxime.caute@cea.fr
Chemla Emmanuel 2 3
Schlenker Philippe 3 4 5
1 Cognitive Neuroimaging Unit, CEA, INSERM, Université Paris-Saclay, NeuroSpin Center , Gif-sur-Yvette , France
2 LSCP (ENS—EHESS—CNRS), Département d’études Cognitives, École Normale Supérieure de Paris , Paris , France
3 Paris Sciences & Lettres Research University , Paris , France
4 Departement of Linguistics, New York University , New York , United States
5 Institut Jean-Nicod (ENS—EHESS—CNRS), École Normale Supérieure de Paris , Paris , France
Vonk Jennifer
Electronic publication date: 2024 Feb 21
Publication date: 2024
Volume: 12
Electronic Location ID: e16800
Received 2023 Jun 22; Accepted 2023 Dec 26
Copyright: © 2024 Cauté et al.
Copyright year: 2024
Copyright holder: Cauté et al.
License: This is an open access article distributed under the terms of the Creative Commons Attribution License, which permits unrestricted use, distribution, reproduction and adaptation in any medium and for any purpose provided that it is properly attributed. For attribution, the original author(s), title, publication source (PeerJ) and either DOI or URL of the article must be cited.
License URL: https://creativecommons.org/licenses/by/4.0/

Keywords: Compositionality, Chimpanzees, Facial expressions, Gestures, Multimodality, Communication, Language evolution, Componentiality, System comparison

Funding: European Research Council (ERC) under the European Union’s Horizon 2020 research and innovation programme 788077 Research was conducted at DEC, École Normale Supérieure—PSL University FrontCog ANR-17-EURE-0017 École Normale Supérieure de Rennes This research was supported by funding from the European Research Council (ERC) under the European Union’s Horizon 2020 research and innovation programme (grant agreement No 788077, Orisem, PI: Schlenker). Research was conducted at DEC, École Normale Supérieure—PSL University. DEC is supported by grant FrontCog ANR-17-EURE-0017. Maxime Cauté is supported by a doctoral grant from the École Normale Supérieure de Rennes. The funders had no role in study design, data collection and analysis, decision to publish, or preparation of the manuscript.

==============================
Using field observations from a sanctuary, Oña and colleagues (DOI: 10.7717/peerj.7623) investigated the semantics of face-gesture combinations in chimpanzees (Pan troglodytes). The response of the animals to these signals was encoded as a binary measure: positive interactions such as approaching or grooming were considered affiliative; ignoring or attacking was considered non-affiliative. The relevant signals are illustrated in Fig. 1 (https://doi.org/10.7717/peerj.7623/fig-1), together with the outcome in terms of average affiliativeness. The authors observe that there seems to be no systematicity in the way the faces modify the responses to the gestures, sometimes reducing affiliativeness, sometimes increasing it. A strong interpretation of this result would be that the meaning of a gesture-face combination cannot be derived from the meaning of the gesture and the meaning of the face, that is, the interpretation of chimpanzees’ face-gesture combinations are non compositional in nature. We will revisit this conclusion: we will exhibit simple compositional systems which, after all, may be plausible. At the methodological level, we argue that it is critical to lay out the theoretical options explicitly for a complete comparison of their pros and cons.

Introduction

Using field observations from a sanctuary, Oña, Sandler & Liebal (2019) investigated the semantics of face-gesture combinations in chimpanzees (Pan troglodytes). The response of the animals to these signals was encoded as a binary measure: positive interactions such as approaching or grooming were considered affiliative; ignoring or attacking was considered non-affiliative. The relevant signals are illustrated in Fig. 1, together with the outcome in terms of average affiliativeness. Two gestures are considered by the authors: Stretched arm gesture (SG), consisting in extending the arm, elbow, wrist and hand straight; and Bent arm gesture, consisting in extending the arm with a bent elbow, wrist and/or hand. Three facial expressions are considered: a Bared teeth face, whereby the signaller reveals his teeth in a grin-like manner, a Hoot face, whereby his lips are funneled and which is often used for vocalizations called Pant-hoot vocalizations; and a neutral face, which we will treat as a lack of facial expression for simplicity. The authors observe that producing non-neutral faces alongside a gesture modifies the responses to the gesture, and most importantly in a non-systematic manner: facial expressions can reduce affiliativeness for some gestures and increase it for others. These observations lead the authors to propose that the meaning of a gesture-face combination cannot be derived from the meaning of the gesture and the meaning of the face. In other words, the interpretation of chimpanzee face-gesture combinations would be non compositional in nature.

Figure 1 Potential combinations of faces and gestures that were considered by Oña et al.

This figure is a modified version of a figure originally published in Oña et al. (DOI: 10.7717/peerj.7623) under a CC-BY license.

We will revisit this conclusion: we will exhibit simple compositional systems which, after all, may be plausible. At the methodological level, we will argue that it is critical to lay out the theoretical options explicitly for a complete comparison of their pros and cons. This new take on their data is motivated by the significant importance of the question of compositionality in primates for evolutionary linguistics. In fact, compositionality is a central property of human language and knowing to what degree it exists in close primate species will allow us to better understand how, and with what step, it emerged. We believe that the data collected by Oña et al. provides a fertile ground for the tools of formal linguistics and that both fields would benefit from being brought closer together. In the section “Compositional vs. Holistic systems”, we will provide working definitions of our classes of systems, and argue why we stuck to this classical dichotomy. In the section “Analysis of the data to be accounted for”, we will report the main observations that our systems will have to account for, mixing observations reported by Oña et al. and new ones obtained after reanalyzing the dataset. In the section “Three possible systems”, we will propose and analyze three systems, compositional or not, that aim to account for our target observations. Finally, we will summarize our results in the “Discussion” section, highlight limitations and propose extensions of our approach.

Compositional vs. holistic systems

Our approach revolves around semantic systems. Formally, semantic systems consist of a set of signals—a lexicon—and a mapping from each signal to one or several meanings—the semantics. In our case, the lexicon is common to all our systems and encompasses Hoot and Bared teeth facial expressions, Stretched and Bent gestures, and any resulting face-gesture combination. In turn, the semantics can be broken into two components, which should be specified by any full semantic system: Meanings for individual gestures, and individual facial expressions

Meanings for each combination (sequence of gestures, sequence of facial expressions, or combinations of gestures and facial expressions)

We will categorize semantic systems based on the relation between (1) and (2). In compositional systems, (2) is derived from (1) and some general rules of composition1 . On the contrary, and as proposed by Oña, Sandler & Liebal (2019), in other systems (2) can also be completely independent from (1). Semantically, the systems making up this latter category are known as holistic systems, since the meaning of the whole is independent from the meaning of the parts.

There are a few terminological considerations that we want to stress here. First, we use a definition of ‘holism’ that is different from Oña et al.’s, who consider that holistic systems’ signals must be indivisible wholes; since we are interested in the semantics of the system, we ask whether the meanings of the signals are indivisible wholes. The distinction between semantically holistic systems with complex signals in their inventory (called componential systems by Oña et al.), and those without, are in fact not relevant to the main aspects of this work. Our aim was to defend that compositional systems may still be compatible with Oña et al.’s data. More generally, we also want to stress that the terms we use here (meaning, lexicon, semantics) are understood under their theoretical meaning in the field of formal languages. Those terms apply in a wide range of situations (e.g., Morse can be understood as a formal language). They also do not bear any connotation whatsoever about a relatedness between the formalized communication system (chimpanzee face-gesture combinations) and human languages.

The category of a semantic system entails different levels of constraints when aiming to propose it: in compositional systems, (1) can be set freely, and (2) is necessarily derived from (1) (given the rules of composition); on the contrary, in holistic systems, both (1) and (2) can be set freely. Compositional systems are therefore more constrained, i.e., more parsimonious, and also likely to be more interpretable. Their final plausibility, however, still depends on the plausibility of the composition rule itself, for the particular species in question.

Beside the lexicon and the compositional rules, systems may rely on pragmatic or enrichment rules. For instance, plausible urgency considerations may require that some pieces of information come early in a sequence, and this may influence the way a message is interpreted if this rule is violated (Schlenker et al., 2016; Narbona Sabaté et al., 2022).

Analysis of the data to be accounted for

Our aim in this paper is to provide a system that can explain the chimpanzees’ responses reported in Oña et al. In Fig. 2, we present these very same data (gratefully made available), with slight modifications in the presentation. Like they do, we show the percentage of affiliative responses to different signals. Going beyond the initial report, here we split the observations depending on the rank of the addressee. Only observations from positive contexts are kept (very few came from negative contexts) and only observations where face, gestures, rank, and response were all encoded (to allow for the new split analysis).

Figure 2 Percentage of affiliative responses to different combinations of gestures and facial expressions from Oña et al.’s raw observations.

Observations were split depending on whether the recipient (the one providing the response) was the subordinate or the dominant.

From Fig. 2, three observations emerge that must be accounted for: Affiliative responses are lower in dominants, especially for Stretched gestures (a new observation obtained from the new analysis by dominance);

Hoot faces tend to decrease affiliative responses for both gestures;

Bared teeth faces increase affiliativeness for Stretched gestures, but decrease affiliativeness for Bent gestures; this is the original puzzling result, and it is especially true for dominant recipients.

These three observations will be more detailed below, and will also be completed by minor observations to support our arguments. In line with Oña et al., we used mixed-effects models. All our models shared the same random effects (sex and identity of both the initiator and the recipient, their dyad, their group and the year of collection of the data). A summary of all models is in the Supplemental Material, and the script of analysis is available at https://doi.org/10.17605/OSF.IO/3P9EB. Unless specified otherwise, our dependent variable was the affiliativeness of the response, encoded binarily as per Oña et al.’s data (resulting in a logistic model). The fixed effects are specified in the text for each model individually. Most models resulted in singular fit, limiting their predictive power; we still decided to include them for completeness.

Three possible systems

We propose below a list of three systems that may account for the chimpanzees’ responses. These systems should be evaluated based on their relative empirical adequacy and a priori simplicity. A summary of all these systems can be found in Table 1, and their evaluation in Table 2.

Table 1 Description of the three systems under consideration.

	System 1
Holistic system	System 2
Trivial compositionality	System 3
Trivial compositionality
+ pragmatic strengthening	
Lexicon	[Bent] = begging (Graham et al., 2018)
[Stretched] = approach (Hobaiter & Byrne, 2011; Graham et al., 2018…)
[Hoot] = antagonism (Nishida, 1983)
[Bared] = submission (Kim et al., 2022)
+ Meanings for complex signals must also be specified arbitrarily	[Bent] = begging
[Stretched] = approach
[Hoot] = antagonism
[Bared] = submission	[Bent] = begging
[Stretched] = approach
[Hoot] = antagonism
[Bared] = fear (Van Hooff & Preuschoft, 2003; Waller & Dunbar, 2005…)	
Compositional rule	None	[A B] means
[A] and [B]	[A B] means
[A] and [B]	
Enrichment rule	None	None	If A is entailed by B, then
[A]prag = ([A]literal and not
[B]literal)	
Interpretation of complex signals (incl. enrichment)	Can be specified arbitrarily	[Bent+Hoot] = begging+antag
[Bent+Bared] = begging+subm
[Stretched+Hoot] = approach+antag
[Stretched+Bared] = approach+subm	[Bent+Hoot] = begging+antag
[Bent+Bared] = begging+fear
[Stretched+Hoot] = approach+antag
[Stretched+Bared] = approach+fear of you (submission)	

Table 2 Qualitative comparison of the three systems under consideration.

	System 1
Holistic system	System 2
Trivial compositionality	System 3
Trivial compositionality
+ pragmatic strengthening	
Pros	Empirically, holistic systems can be made to fit all observations	Fewer arbitrary lexical rules
Trivial compositionality is most natural	Fewer arbitrary lexical rules
Trivial compositionality is most natural
Enrichment mechanisms have been proposed elsewhere
Fear, an emotion, is natural as a meaning for a facial expression	
Cons	Loosely constrained
Use of Bared by dominants is at odd with a submission meaning (can be changed without consequences)	Use of Bared by dominants is at odds with a submission meaning	Enrichment mechanisms may be seen as complex	

System 1—holistic/componential system

Oña et al. suggest that a holistic system2 accounts best for their data. What holistic system? Well, there are virtually no constraints on the meaning of combinations: in essence, holistic systems are a restatement of the list of correspondences observed between contexts and signals, be the signals simple or complex. Such a system may thus be seen as provided by Table 1 above, and Oña et al. indeed do not propose a more explicit system.

We will nonetheless cover some of the evidence for what the meanings of simple signals may be, as these are the parameters of the theories that are common between holistic and compositional systems.

Isolated Gestures. Oña et al. focused on one-armed static gestures, which they broke down into two categories, with either a fully extended arm (stretched arm gesture, hereafter Stretched) or an arm whose wrist or elbow was bent (bent arm gesture, hereafter Bent). The authors related them to pre-existing categories in the literature (Hobaiter & Byrne, 2011; Roberts, Vick & Buchanan-Smith, 2012). Stretched gestures were assimilated to ‘Reach’ gestures from the literature, thought to convey a meaning of begging for some food or object (Oña, Sandler & Liebal, 2019; Hobaiter & Byrne, 2011; Pollick & de Waal, 2007; Hobaiter, Byrne & Zuberbühler, 2017); Bent gestures were related to ‘Wrist offer’ gestures, thought to express some form of submission (Oña, Sandler & Liebal, 2019; Hobaiter, Byrne & Zuberbühler, 2017).

This view can be further refined. First, although Reach/Stretched indeed often results in food/object acquisition (Graham et al., 2018), other outcomes also appear satisfactory to the signaller, such as giving away food (Hobaiter & Byrne, 2011), but also moving closer and initiating various contacts (Graham et al., 2018), or initiating mother-child joint travel (Fröhlich, Wittig & Pika, 2016). We thus suggest that the meaning of Stretched is more general: it requests the receiver to approach.

Second, the analysis of Bent may be complex. The coding scheme of Oña et al. for Bent indeed encompasses Wrist offer, but it also includes other gestures with a bent hand or elbow (for a total of 22 distincts sets of features). This includes in particular the gesture ‘Arm raise’, where the elbow is bent with the forearm raised vertically (see Graham et al., 2018) for an illustration). These two versions of Bent, Arm raise and Wrist offer, seem to occur in different contexts (Liebal, Call & Tomasello, 2004); and even for Arm raise alone, no clear common component emerges from apparently satisfactory outcomes (which include moving closer or away, and stopping current behavior; Graham et al., 2018). To settle for a single meaning and fit Oña et al.’s framework, we suggest that Bent be associated with a begging meaning (asking e.g., for food as in Graham et al. (2018), but also possibly for assistance).

To summarize, we propose the following meanings for gestures: – Stretched is a request for the receiver to approach (to, e.g., give food away (Hobaiter & Byrne, 2011), initiate contacts (Graham et al., 2018) or mother-child joint travel (Fröhlich, Wittig & Pika, 2016))

– Bent is used for begging (asking, e.g., for food (Graham et al., 2018), but also possibly for assistance)

These meanings are not the most typical ones in the literature, but they are simple and they can explain the data from Oña et al., including the new observation (A), that is that dominance tends to decrease affiliative response, especially for Stretched gestures. In the case of gestures without a specific facial expression, this is illustrated by the left panel of Fig. 2. Dominance clearly plays a role in deciding the outcome of an interaction: non-dominants were much more likely to react in an affiliative fashion than dominants. In fact, 72% of all the non-dominants’ responses under any expression were affiliative, against 29% of those of the dominants’. A GLMM analysis confirmed this finding (β = −1.37; z = −2.35; p = 0.01), indicating that dominants were exp(1.37) = 3.9 times less likely to react in an affiliative fashion. More crucially, with neutral faces, the two gestures Bent and Stretched differed when directed towards dominants: Bent did not display any bias (50% of responses are affiliative; GLMM restricted to neutral bent: βintercept = 0.035; z = 0.03; p = 0.976) when Stretched was markedly non-affiliative (22%; βintercept = −11.13; z = −2.3; p = 0.02).3

These differences are compatible with the proposed meanings above. In fact, asking a dominant to approach is somewhat doomed to fail. Conversely, begging for help from a dominant may sometimes be met with success, as the dominant may be in a position to provide help. We would also expect dominants to beg for help less often than non-dominants, that is to produce fewer (isolated) Bent gestures. This prediction is met in Oña et al.’s data: only eight Bent gestures originated from dominants, against 20 from non-dominants. This difference could not be explained by a scarcity of observations from dominants, as 21 (isolated) Stretched gestures were observed in dominants against 18 in non-dominants. A GLMM analysis further confirmed that an isolated gesture was 3.2 times more likely to be a Bent gesture if it originates from a non-dominant, although it was statistically only a trend (β = 1.17; z = 1.8; p = 0.07).

Facial expressions. Facial expressions in isolation were not recorded by Oña et al. However, we can try to turn to the literature to get a better idea of what meanings are plausible. Bared teeth displays (a teeth-displaying grin, also known as “fear grins”) are widely reported in the literature and are mostly associated with submissive behaviors (Kim et al., 2022)4 . Hoot (a facial expression with funneled lips) is often associated with pant-hoot vocalizations although sometimes occurring silently. Pant-hoots are used in various situations, especially those involving general excitement (Parr, Cohen & de Waal, 2005; Notman & Rendall, 2005): distress or bluff displays (Parr, Cohen & de Waal, 2005), threatening (Nishida, 1983) or arrival at food sources (Notman & Rendall, 2005; Clark & Wrangham, 1994). They are indeed often associated with pilo-erection (Parr, Cohen & de Waal, 2005; Notman & Rendall, 2005). However, they also appear to serve during travel, either to space out neighboring, less numerous groups (Wilson, Hauser & Wrangham, 2001) or to maintain contact with nearby allies or associates within a single group (Mitani & Nishida, 1993). Some authors argued that they, too, could trigger arousal (e.g., associated with social separation) (Notman & Rendall, 2005). It could, on the other hand, also be that pant-hoots serve a general purpose of revealing identity and/or social status, as they are generally produced by high-ranking males (Clark & Wrangham, 1994). However, it must be noted that most of these usages rely on the vocal components of the pant-hoot: in particular, pant-hoots are much more frequently produced when associates are within earshot (compared to other nearby control males) but not when the same associates are already present alongside the caller (Mitani & Nishida, 1993). Since we only consider facial expressions here, we may want to focus on situations that more specifically rely on visual contact to infer a general meaning for Hoot faces. These include agonistic situations such as fighting, in which chimpanzees ostensibly pant-hoot against their enemy (Nishida, 1983). We thus propose to assign Hoot faces a general meaning of antagonism for our candidate models.

To summarize, we propose to use the following meanings for now: – Hoot faces show antagonism (Nishida, 1983)

– Bared faces show submission (Kim et al., 2022)

We would like to highlight that these meanings should not be considered as the exact and definitive meanings of Hoot and Bared. Although we would like in an ideal world to be able to fit every single use case of a signal, we have no choice but to settle for a single, explicit meaning to properly compare models. This is what has been done here, in the hope of explaining at least a good part of the cases (especially in the case of visual contact). We also immediately note that Bared has also been proposed to be associated with fear (Parr, Cohen & de Waal, 2005; Preuschoft, 1999; Brent, Kessel & Barrera, 1997; Van Hooff & Preuschoft, 2003), and we will move to this more general option in System 3.

Summary. We detailed the first part of a holistic system which is likely common to other systems: the analysis of isolated signals. We proposed meanings (summarized in Table 2) compatible with both observation (A) extracted from Oña et al.’s data and other results from the literature. Note again that the literature is only moderately supportive of the meanings we assigned to gestures, as the repertoire covered here probably encompasses more than two unique gestures. For complex signals, meanings that match observations (B) and (C) should also be assigned to the 2 × 2 combinations of gestures and facial expressions. This is left implicit here (and in Oña et al.’s original manuscript), as the exercise is not further constrained in a holistic system.

We will now present more constrained systems deriving meanings for complex expressions from compositional rules, with or without further enrichment rules. These enrichment rules are additional pragmatic mechanisms that can specify meanings based on contextual information and/or world knowledge. Unless notified otherwise, we will stick to the meanings presented above for signals in isolation.

System 2—trivial compositionality alone

Holistic systems assume that combinations are assigned their own meaning as if it had no relation to their components. However, when two signals co-occur, say a gesture and a facial expression, a default meaning that may be obtained is the mere sum of their meanings, as if the two signals had occurred for their own reasons. Trivial compositionality, as discussed in Schlenker et al. (2017) for sequences of signals, represents this most simple compositional rule: the meaning of a combination of expressions is the conjunction of the meanings of these expressions.

Simple as it is, trivial compositionality can produce rich effects. Hoot faces, for instance, are expected to convey antagonism, and this should happen whether or not they are combined with a gesture, any gesture. The data matches this expectation (observation (B); middle panel of Fig. 2): Hoot faces overall decrease affiliative responses from 57% of all responses in the case of neutral faces to only 25%. A GLMM analysis on response score by Hoot vs. neutral faces confirms the decrease (β = −1.15; z = −2.1; p = 0.03)5 . One puzzling observation, however, is that Bent gestures are still used in combination with Hoot faces (N = 31 observations). This use may appear unexpected if we assume Bent to signal begging, given how counterproductive it is to signal antagonism while begging for help. However, some authors (Pollick & de Waal, 2007) suggest that facial expressions may be to some degree involuntarily displayed by chimpanzees6 (which, as must be noted, does not devoid them of semantics (Frith, 2009)). An imperfect control over facial expressions could thus explain this suboptimal behavior: animals may need help, but also feel antagonism, which they cannot refrain from showing. Indeed, under an involuntary approach to facial expressions or vocalizations, trivial compositionality is a natural compositional rule: signals add up to one another, as their triggers add up to one another.

Similarly, Bared faces occurring alongside with a gesture are expected to be independent signals of submission. While we lack data points to analyze this hypothesis in depth, it seems that it shifts dominant responses to some extent (Observation (C); Fig. 2, right panel). In fact, only 11% of dominants’ responses to Bent+Bared are affiliative, against 50% in the baseline, isolated Bent situation (GLMM on Bared vs. neutral faces for Bent gestures: β = −2.17; z = −1.7; p = 0.09). Under our previous assumptions, this effect is coherent with signals of submission: if helping plays a role in asserting dominance, it becomes unnecessary (costly) if this dominance is clear already. Like for hoot faces, the use of this counterproductive Bent+Bared combination could stem from a poor control over facial expressions. Unlike responses to Bent+Bared gestures, the proportion of affiliative responses to Stretched+Bared skyrockets to 75% from 22% for isolated Stretched (β = 22; z = 2.4; p = 0.01). Again, a submission meaning is here insightful: asking to approach while signaling submission is likely to forecast an offering, in which case it would be interesting to approach.

There is also a problematic prediction to be discussed here. If Bared signals submission, dominants should not use this facial expression. Yet n = 10 occurrences of a dominant signaling a Bared face were reported (also observed in Kim et al. (2022), Preuschoft (1999)).

Summary. This (trivial) compositional system thus meets most of the empirical targets. Importantly, it accounts for the original remark that there seems to be inconsistent variation in the application of the facial expressions to various gestures. System 2 thus provides a proof of concept for a compositional explanation of Oña et al.’s main challenge. However, it comes with a problem due to the mere lexical meaning of Bared, which will be addressed in System 3 that follows.

System 3—trivial compositionality with the informativity principle

System 3 is a variant of System 2, with two main changes. First, the Bared teeth face is assumed to signal fear, rather than submission. In a sense, fear can still indicate submission when it is directed toward the interaction partner himself (Brent, Kessel & Barrera, 1997). At the same time, it is not bound to submission as it can also be directed toward an external threat. This fixes the main problem with System 2: the use of Bared by dominants is not a signal of submission, but a signal of pure fear, presumably fear of an external threat, given the context (dominants are usually not afraid of their subordinates). On the opposite, the use of Bared by subordinates also signals fear, but now presumably of the dominant interaction partner; it can thus still be interpreted in this case as a signal of submission and yield the same effects as in System 2. Several observations from the literature support this proposed meaning to a certain degree. First, in other species Bared teeth displays seem to be used as a display of fear, especially in submissive interactions of despotic species (Van Hooff & Preuschoft, 2003). In fact, even when transposed to human faces (correspondences being driven by facial musculature), human subjects interpret these displays as expressing fear (Parr & Waller, 2006). For chimpanzees specifically, the expression of fear is consistent with many occurrences of Bared, such as aggression (Preuschoft, 1999; Van Hooff & Preuschoft, 2003; Waller & Dunbar, 2005). However, some authors proposed that it could also display a more generic meaning of anxiety (Van Hooff & Preuschoft, 2003), or even benign intent (Waller & Dunbar, 2005).

Second, System 3 also comes with an additional pragmatic rule of interpretation, known as the Informativity Principle (or Pragmatic Strengthening, see Schlenker et al., 2016): If a signal is in competition with a more specific, alternative signal, its use is restricted to what isn’t covered by said alternative. This mechanism may apply to Stretched+Bared here; we schematize the different steps in Fig. 3, and explain it here in prose. Stretched+Bared comes out as an approach request (Stretched) and a signal of fear. By comparison, Bent+Bared is here assumed to be a begging request (Bent) and a signal of fear (Bared). This latter meaning can be seen as more specific, if a begging request is seen as a request to approach and to help. Hence, by the informativity principle, Stretched+Bared ends up being interpreted as a request to approach, but not for help, in a situation of fear. Here, the fear is presumably not a fear of an external threat, because then help would be welcome. Hence, it is presumably a fear of the addressee, that is a signal of submission. Overall then, a combination of factors conspire: (i) the application of the Informativity Principle (ii) the recognition that fear is either fear of the addressee (submission) or fear of an external threat, (iii) that begging can be decomposed as both a request to approach and a request for help, (iv) that a request not for help is incompatible with an external threat. Skipping over the details, the net result is that the submission meaning of the Stretched+Bared combination obtains again here, despite the fact that submission is not directly encoded in the Bared facial expression anymore.

Figure 3 Schematic layout of the derivation of the final interpretation of the combinations Bent+Bared and Stretched+Bared in System 3 with pragmatic reinforcement.

The above reasoning explains Observation (C). The reader can verify that, since other elementary meanings and principles were kept from System 2, System 3 also accounts for the Observations (A) and (B) (mostly because they did not rely on the meaning of Bared, though informativity competition effects must be checked for).

Summary. System 3 assigns Bared an emotional literal meaning (fear) rather than a relational one (submission). This solves the problem of System 2, that it is used by dominants. Still, the otherwise classical meaning of submission can be retrieved through the Informativity Principle, in the case of a combination with the gesture Stretched. Both the new meaning (Kim et al., 2022) and this additional pragmatic mechanism (Schlenker et al., 2016) have been independently proposed in the literature.

We finally want to highlight the appeal of an emotional literal meaning for Bared. Not only is this meaning extremely simple, it also is consistent with how emotions are signaled also in humans through facial expressions (see, e.g., Frith, 2009). Furthermore, assigning emotion meanings to facial expressions7 may offer a simple justification for a trivially compositional system, as discussed above: facial expressions reflect the emotional state of the signaller, which piece of information can be added to the message conveyed.

Conclusions

Oña et al. proposed that holistic (i.e., non-compositional) analyses best explain chimpanzees’ face-gesture combinations. In this work, we exhibited and compared three possible systems for these combinations. The first system is holistic, independently stipulating meanings for each signal, whether a simple or a complex signal. The second and third systems implement a basic form of compositionality called Trivial compositionality (Schlenker et al., 2017). This rule is natural in perception (stating that the meanings of every signal received accumulate) and in production (stating that all signal triggers generate a signal). The third system is further enriched through a pragmatic strengthening mechanism (Schlenker et al., 2016).

All three systems account for the data gathered by Oña et al. (observations (A), (B), (C)) while being compatible with previous literature. However, they differ in their Degrees of Freedom (DF): holistic System 1 involves 6 DF in the form of lexical stipulations (one per gesture and combinations8 ), while compositional systems only involve four DF (one per isolated signal, if compositional rules are given). As such, holistic systems are less principled, and thus a priori less convincing. The situation would be even more extreme once a couple more single signals are taken into account, and create many further combinations. Furthermore, System 3 relies on very elementary emotional meanings for isolated facial expressions. As a result, the compositional System 3 is a very powerful explanation, and a clear challenger to a less constrained holistic system.

Empirically, let us describe two interesting extensions of the present investigation. First, we only considered two distinct gestures Stretched and Bent, as per Oña, Sandler & Liebal (2019). The more chimpanzees’ manual gestures a compositional system could integrate, the more convincing it becomes (as the order of magnitude of its DF would be linear in the number of gestures, while it would explain many more combinations). Second, full gestures may be analyzed as combinations of more elementary movements (see, e.g., Roberts et al., 2012, and the initial descriptions of Oña et al. before clustering). These components may or may not have their own meanings that combine compositionally; e.g., Stretched’s extended arm could call for the approach, while the hand orientation could convey the desired outcome: food-sharing vs grooming. Such analysis has already been proposed for the calls of chimpanzees (Fedurek, Zuberbühler & Dahl, 2016). If elementary components indeed have their own meanings, the combinatorial possibilities and depth of explanations of a compositional system, if sustainable, increase. Third, more multimodal combinations could be considered. For instance, one could add vocal signals to the systems (Hobaiter, Byrne & Zuberbühler, 2017), as they were not registered in the present study. Compositional systems would yield almost immediate predictions there too, showing their full power and how data can challenge them more than holistic systems. In particular, the addition of meaningful vocal components to Hoot faces could, in a compositional system, explain the diversity of uses for pant-hooting. In such a setting, the meaning of antagonism we proposed could already be itself an enriched meaning derived from the context of visual contact with the recipient; the literal, context-free meaning could be even weaker and linked to, e.g., general arousal (Parr, Cohen & de Waal, 2005; Notman & Rendall, 2005).

Methodologically, we suggest that a systematic comparison of best-fit systems involving a range of different mechanisms could significantly help illuminate field data interpretation. Key features of these systems primarily include their fit to empirical data, but also the naturalness of the lexical entries they rely on, the plausibility of their independent general assumptions (e.g., pure memory storage, trivial compositionality, informativity principle) and their resulting predictive power (represented by a low number of degrees of freedom). Oftentimes, the empirical data is sparse and can be fit with various kinds of such systems; constrained systems typically enjoy higher explanatory force, at the expense of more sophisticated mechanisms, some of which like trivial compositionality may be highly natural.

Supplemental Information

Supplemental Information 1 Summary description and output of all GLMER models.

‘Model’ describes the dependent variable (left of the tilde ‘~’) and the fixed effects (right of the tilde ‘~’, “1” means intercept only). ‘Filters’ describe the subset of the data on which the model was fitted and ‘Estimates’ report the estimates and p-values of the intercept and regression coefficient when suited. All details (e.g., random structure) are available in the R script at https://osf.io/gny5k.

This work would not have been possible without the work of Oña et al. and its publicly available data. We would also like to thank Camille Coye, Maël Leroux, Salvador Mascarenhas, Pritty Patel-Grosz and Benjamin Spector for discussion and feedback and the reviewers and academic editor.

Additional Information and Declarations

Competing Interests

Author Contributions

Data Availability

1 The definition for compositionality here and in Oña et al. is the standard one: “The meaning of a complex expression is a function of the meanings of its constituents and the way they are combined.

2 Oña et al. call such systems “componential” to highlight the fact that the form (but not the meaning) of some signals are combinations of elementary signals.

3 This diverges from Oña et al.’s analyses, where both gestures come out as actually affiliative. This difference stems from the possibility here to factor out the independent role of dominance: as responses to dominants are overwhelmingly affiliative, the average answer remains somewhat so.

4 We will cover in more depth this facial expression in the presentation of System 3 which revises this associated meaning.

5 Significance survived the addition of gestures and/or dominance as fixed effects, or the interaction between the two. Since neither added factor was significant, we report the results without these effects.

6 Actually facial expressions may be involuntarily displayed by humans too: frowning when seeing someone/something you dislike is a fairly natural, reflex reaction.

7 Note that, although we did keep the relational meaning of antagonism for Hoot, this expression could rather be assigned an emotional anger meaning. Akin to fear being read as submission, antagonism could then be seen as the dyad-internal expression of anger. This also offers a new possible direction to refine the interpretation of Hoot: anger could be against the recipient or against something external. If so, Bent+Hoot could be interpreted as asking for help against something external angering, rather than help despite antagonism.

8 We did not have data for facial expressions in isolation.

The authors declare that they have no competing interests.

Maxime Cauté analyzed the data, prepared figures and/or tables, authored or reviewed drafts of the article, contributed to designing candidate models from the data - Category 1, and approved the final draft.

Emmanuel Chemla analyzed the data, prepared figures and/or tables, authored or reviewed drafts of the article, contributed to designing candidate models from the data - Category 1, and approved the final draft.

Philippe Schlenker analyzed the data, prepared figures and/or tables, authored or reviewed drafts of the article, contributed to designing candidate models from the data - Category 1, and approved the final draft.

The following information was supplied regarding data availability:

The R code of reanalysis is available at OSF: https://doi.org/10.17605/OSF.IO/3P9EB.

Our analysis code is available at OSF: Chemla, Emmanuel, and Maxime Cauté. 2024. “Inconsistent Effects of Components as Evidence for Non-Compositionality in Chimpanzee Face-Gesture Combinations? A Response to Oña et al. (2019).” OSF. January 8. 10.17605/OSF.IO/3P9EB.

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
