# Peer review of "Inconsistent effects of components as evidence for non-compositionality in chimpanzee face-gesture combinations? A response to Oña et al (2019)"

_PeerJ, doi:10.7717/peerj.16800_

## Round 0.1 · original submission · Major Revisions

Based on the insightful comments of three very expert reviewers, I am afraid I cannot accept your commentary in its current form. It seems that your commentary involves a misinterpretation of the original authors' argument and that your re-analysis may also introduce some misleading information. All of the reviewers and I see the value in re-analyzing original work and providing alternative interpretations. I would encourage you to take the reviewers' comments to heart and consider how to build upon the important work you have done. If you can substantially revise the manuscript to take their concerns into consideration, I would be happy to reconsider its merits.

Reviewer 1 ·

Basic reporting

See 4

Experimental design

See 4

Validity of the findings

See 4

Additional comments

I have reviewed the paper by Caute et al. This paper aims to revisit the findings of Ona et al. (2019) which conclude that face-gesture combinations in semi-captive chimpanzees represent a holistic system where the meaning of the whole is not related to the meaning of the parts. Caute et al. challenge this conclusion instead stating that a compositional analysis (particularly one enriched by a pragmatic strengthening mechanism) better captures the data.

I found this to be an interesting and constructive contribution to the findings of Ona et al (2019) but also to the ongoing debate regarding the semantics of non-human animal combinations more generally. Nevertheless, I have comments and suggestions that it would be great if the authors could consider and address.

1. L150: The authors run a “two-way chi square” but don’t really provide any further details on the statistical methods or reanalysis approach. I appreciate this is a comment on a previous paper but given there is new analysis here I think some more details would be useful. Furthermore, I was also wondering if this is really the right analysis here since the data seem to suffer from issues with pseudoreplication (repeated measures) for which more modelling-based analyses would be more appropriate. Can the authors comment on this?
2. L49: Since some readers will not be so familiar with the Ona et al. (2019) paper and to avoid readers having to go back and forth, I think it could be helpful to reiterate how compositional semantic systems are defined and also provide working definitions for other terms including “semantics”, “holistic” etc
3. L63: change to “in question”
4. L108 “dive some more” sounds a little colloquial. Perhaps change?
5. L184: Can the authors elaborate on what they mean by enrichment rules?
6. There is also some inconsistency in the terminology used: sometimes “display” is used, other times “signal” or “meaning”. If the authors continue to use different terms here it might be useful to clarify what the differences are.
7. I was wondering how this critique here fundamentally differs from what is already forwarded in the recent paper by Schlenker et al. (2023)? The competing hypotheses and the overall conclusions appear to be superficially quite similar.

·

Basic reporting

In their response article "Holism vs. compositionality in chimpanzee face-gesture combinations, a response to Oña et al. (2019)", the authors re-analyzed data presented in Oña et al. 2019 that contained a data sample from chimpanzees in a sanctuary. The chimps were observed regarding the usage of multimodal compositional structures in their communicative system (face and gesture combinations). Compositional structures, as conceptualized in this context, are those in which individual gestures and facial expressions possess distinct meanings when employed independently, and when combined, they generate novel meanings that are derived from the meanings of their parts. Oña et al. (2019) identified that, in positive social contexts, certain combinations of face and gesture had an impact on the probability of eliciting specific responses from recipients during dyadic interactions. However, due to the inconsistent nature of these responses, the authors cautiously interpreted their findings, suggesting the potential presence of compositional structures within the communicative system of chimpanzees. We believe that our data did not provide substantial evidence for compositionality per se. However, based on consistent responses to certain face-gesture combinations, we were able to conclude that the data represent a preliminary stage in the progression toward compositionality, which we termed "combinatoriality."

In their re-analyses, Cauté et al. used the same data as a basis and revisited our conclusion. Whereas I believe that it is a crucial step to re-analyze (and also replicate) existing data, and it is not done enough yet as the academic landscape often inhibits these tasks due to the need to publish new and exciting findings, I am very grateful that the authors took time and effort and take this vital step. However, although claiming to have found compositional structures in chimpanzee communication, I have serious concerns regarding their claims.

To begin with, the title of their response article is puzzling, since it contains a dichotomy between holism vs compositionality, a dichotomy that we rejected. As outlined in our paper, we do not claim that chimpanzees use a holistic system in their face+gesture combinations. In fact, in our conclusion, we state: "But neither were the facial and manual components that we tracked conveyed and interpreted holistically, i.e., as indivisible wholes. Intriguingly, we found something in between—the ability to recombine parts of a visual display resulting in different effects on conspecifics in particular contexts—a property that we refer to as componentiality."

Another problematic issue is that the re-analysis is based on assigning a meaning to the single unit, the "hoot face" (i.e., hoot faces show antagonism," L163). Although the authors mention that "… the literature provides meanings for hoot (a facial expression with both funneled lips, often associated with pant-hoot vocalizations although sometimes occurring silently)...”, no reference is provided for this claim, and when going into the literature, one can find that pant hoots (hoot face with co-occurring vocalization) do seem to have contradictory meanings depending on within- or between-group usage. The signals seem to function in spacing neighboring groups (Wilson et al. 2001) and maintaining cohesion among individuals within groups (Goodall 1986; Mitani & Nishida 1993). Since Oña et al. 2019 recorded data from within group interactions, one could instead, assign the meaning "affiliation" to the hoot face, and not "antagonism" Cauté et al. claim. Overall, pant hoots seem to correspond to a variety of functions, depending on the context in the groups. In fact, we noted in the "Coding" section: "Pant-hoots occur in various situations that include general excitement, such as distress or bluff displays (Parr, Cohen & De Waal, 2005; Van Hooff, 1973), as part of ritualized agonistic displays of adult male chimpanzees (Mitani et al., 1992), or upon arrival at an abundant food source (Notman & Rendall, 2005), suggesting that it might have various functions." We were hoping to disentangle meaning by its compositional usage. Therefore, assigning a strict meaning to the hoot face, on which the re-analyses of Cauté et al. is based, is premature, and it is therefore impossible to subscribe to the subsequent arguments in their article.

If these issues could be resolved and the authors in Cauté et al. could make a case supporting compositional structures in chimpanzee face+gesture combinations, I would be glad to revisit the results.


Minor comments:
L 62: (add) also likely to be more interpretable/communicative
L 233: I don't think any chimp gesture can be considered 'lexical' at this point: delete 'lexical'


Another problematic issue is that the whole re-analyses is based on assigning a meaning to the single unit "hoot face" (i.e., hoot faces show antagonism", L163). That assignment of meaning is not supported by any reference although the authors claim that "However, the literature provides meanings for hoot (a facial expression with both funneled lips, often associated with pant-hoot vocalizations although sometimes occurring silently), ...". But no reference is provided and when actually going into the literature, one can find that pant hoots do seem to have contrary meaning depending on within or between group usage. They do seem to function in spacing neighbouring groups (Wilson et al. 2001) and also maintaining cohesion among individuals within groups (Goodall 1986; Mitani & Nishida 1993). Since Oña et al. 2019 recorded data from within group interactions, one could - if any- rather assign the meaning "affiliation" to it and not antagonism as used in Cauté et al. Overall, pant hoots seem to own a variety of functions, depending on the context in the groups. It might not have been the best pick for the original analyses in Oña et al. 2019, but since it was also recorded in which context the signal appeared, meaning assignment might be possible, although Oña et al. 2019 did not have data on single usage of the facial expression "hoot". They do explain in the "Coding" section: "Pant-hoots occur in various situations that include general excitement, such as distress or bluff displays (Parr, Cohen & De Waal, 2005; Van Hooff, 1973), as part of ritualized agonistic displays of adult male chimpanzees (Mitani et al., 1992), or upon arrival at an abundant food source (Notman & Rendall, 2005), suggesting that it might have various functions." Therefore, assigning a strict meaning to it were the rest of the re-anaylses of Cauté et al. is based on seems too premature, and it is therefore impossible to follow the subsequent arguments in their article.

An additional concern arises regarding the reanalysis as it hinges on attributing a specific meaning to the single unit "hoot face" (referred to as hoot faces showing antagonism, L163). However, this assignment of meaning lacks supporting references, despite the authors' assertion that "the literature provides meanings for hoot" (a facial expression characterized by funneled lips often associated with pant-hoot vocalizations, occasionally occurring silently). Nonetheless, no citations are provided, and the literature reveals that pant hoots indeed appear to have contrasting meanings depending on whether they occur within or between groups. They serve to create distance between neighboring groups (Wilson et al., 2001) while also fostering cohesion among individuals within groups (Goodall, 1986; Mitani & Nishida, 1993). Since Oña et al. (2019) recorded data from within-group interactions, one could arguably assign the meaning of "affiliation" to pant hoots rather than the antagonism suggested by Cauté et al. overall. Pant hoots appear to have a range of functions contingent on group context. While it may not have been the optimal choice for the original analyses in Oña et al. (2019), the inclusion of contextual information allows for the potential assignment of meaning, despite the absence of data on the solitary usage of the facial expression "hoot" in Oña et al. (2019). As explained in the "Coding" section, pant hoots occur in various situations, including general excitement such as distress or bluff displays (Parr, Cohen & De Waal, 2005; Van Hooff, 1973), as well as ritualized agonistic displays among adult male chimpanzees (Mitani et al., 1992), or upon arrival at an abundant food source (Notman & Rendall, 2005), suggesting their multifunctionality. Consequently, prematurely ascribing a definitive meaning to this expression, upon which the rest of Cauté et al.'s reanalysis is based, hinders comprehension of the subsequent arguments presented in their article.


Minor comments:
L 62: (add) also likely to be more interpretable/communicative
L 233: I don't think any chimp gesture can be considered 'lexical' at this point: delete 'lexical'

Experimental design

no comment

Validity of the findings

no comment

Additional comments

no comment

·

Basic reporting

no comment

Experimental design

no comment

Validity of the findings

no comment

Additional comments

In this paper the authors re-analyzed data of face-gesture combinations of chimpanzees published by Oña et al. 2022. By re-analyzing the original data and including also the dominance status of the sender and receiver, the authors suggest three systems in which two of them indicate a compositional system of face-gesture combinations instead of a holistic system. The authors conclude that it is important to lay out first theoretical options explicitly for a complete comparison of pros and cons for a holistic or compositional system.
First of all, according to my reading, Oña et al. 2022 did not explicitly claim that their data suggest a holistic system. They concluded that their data may indicate “componentiality, a necessary prerequisite for compositionality, a prerequisite which has not been identified as such in previous work.” (first paragraph, page 17/23). Therefore, I don’t understand the claim that Oña et al. 2022 suggest a holistic system. Please, elaborate in more detail why you think that Oña et al. 2022 suggested a holistic system. Moreover, as interesting and valuable the given approach of discussing different compositional systems is, I have some concerns about the interpretation of the meaning of the different gestures and facial expressions, which I outline in detail below.


L45: Please, explain briefly the gesture-face expressions in the text because readers not familiar with the original paper may struggle to follow your argumentation.
L84: Not clear, in combination with the stretched arm gesture? Please, explain in more detail.
L95: Please add references from which you derived the lexical meaning and add the lexical meaning of “Hoot”
Figure 3: Please, add references indicating the potential meaning of bent + bared = approach and help due to an external threat. As far as I understood the Oña et al. 2022 paper, they did not indicate that these gestures and facial expression were given in response to an external threat
L112: This should be explained already earlier, see comment above
L167: Please, explain why you consider hoot-faces as antagonism; the fact that they are also produced in combination with pant-hoots does not justify antagonism because pant-hoots are given in many different contexts.
L241: Please, explain why you assume that the bared teeth face is now signaling fear. I don’t think that submission can be equaled with fear because submissive signals are usually ritualized signals that can indicate either the immediate behavioral response “submission” or subordination as a behavioral state, with the latter one not necessarily being associated with a state of fear.
L243-245: This sentence contradicts the sentence in lines 241-243. In the first sentence you argue that bared may indicate fear of an interaction partner (interlocutor) and not in response to an external threat, whereas you claim in the following sentence that bared may indicate fear of an external threat.
L304: I would recommend to remove the word “brute”

---

## Round 0.2 · Minor Revisions

The reviewers are happy with your revision, except for some minor comments regarding the statistical model as identified by Reviewer 3. Please make this final change and the edits listed below so that I can formally accept the paper.

Please remember to complete your acknowledgements.
Please add "the" in front of "section" on lines 64 and 66, 68.
Line 74 "consist in" should be "consist of"
Line 91, "which" should be "that"
Please add , after i.e. and e.g. throughout
The word "only" is misplaced on lines 93 and 94.
Line 119, "this data" should be "these very same data"
Lines 298-299, either "an independent signal" or delete the "an" and leave plural.
Line 309, "a tribute"??

Reviewer 1 ·

Basic reporting

NA

Experimental design

NA

Validity of the findings

NA

Additional comments

Pending.

·

Basic reporting

No comment

Experimental design

No comment

Validity of the findings

No comment

Additional comments

Thank you very much for considering all points raised in the first round of revision. The arguments are now outlined much clearer and easier to follow. I understand that we do have a different conceptualisation of what a "holistic system" is and why you would not agree with our interpretation of the data. I apologise, but I still do not feel comfortable with the following sentence in the abstract: "So much so that they propose that the meaning of a gesture-face combination cannot be derived from the meaning of the gesture and the meaning of the face, hence arguing for a so-called “holistic” system, and against a compositional system." After having followed all the arguments in the discussion regarding this point, I understand what is meant. However, the reader might understand this in the wrong way and the explanation comes only later in the text to fully understand its scope. It would appreciate it if at least in the abstract the authors would re-phrase the sentence therefore to avoid confusion. We did not find evidence for a strong case of compositionality in our data, this is all we wanted to say. We are not excluding the possibility that chimpanzees are able to produce and perceive compositional structures. Therefore, we welcome the proposed idea of the existence of compositional structure in chimpanzee communication. We believe future collaborations between the different research fields will be succeeding in finding more cases of it.

I agree with all other changes that has been made in the manuscript. I am sure and delighted that this article as a response to ours will spark fruitful and constructive discussions.

·

Basic reporting

The authors provide in their revision a thorough answer to the reviewer’s main cristicism and I’m happy with the revised version. Especially, I’m delighted that the authors explain in more detail the terminology to avoid potential misalignments. I have only one minor comment about the statistcial analyses (see below).

L 169: please explain why you used for affiliativeness a logistic model? Did you include affiliativeness as a binary response, occurring yes or no?
It would also be helpful to include a table with the summary output of the models, so that it is easier to understand to which model the authors are later on referring to.

Experimental design

L 169: please explain why you used for affiliativeness a logistic model? Did you include affiliativeness as a binary response, occurring yes or no?
It would also be helpful to include a table with the summary output of the models, so that it is easier to understand to which model the authors are later on referring to.

Validity of the findings

no comment

Additional comments

no comment

---

## Round 0.3 · accepted · Accept

Thank you for making the final minor edits requested by the reviewers. I think this paper will provoke valuable discussion.